# A Special Case of Relapsing–Remitting Bilateral Encephalitis: Without Epilepsy, but Responding to Rituximab and with a Brain Biopsy Coinciding with Rasmussen Encephalitis

**DOI:** 10.3390/brainsci13010017

**Published:** 2022-12-22

**Authors:** Pei Liu, Xuemei Lin, Shenghua Zong, Yan Yan, Zhongzhong Liu, Qingli Lu, Qiaoqiao Chang, Songdi Wu

**Affiliations:** 1Department of Neurology, The First Hospital of Xi’an (The First Affiliated Hospital of Northwest University), Xi’an 710002, China; 2Xi’an Key Laboratory for Innovation and Translation of Neuroimmunological Diseases, Xi’an 710002, China; 3Department of Psychiatry and Neuropsychology, Division Translational Neuroscience, School for Mental Health and Neuroscience, Maastricht University, 6229 HX Maastricht, The Netherlands; 4Department of Pathology, The First Hospital of Xi’an (The First Affiliated Hospital of Northwest University), Xi’an 710002, China

**Keywords:** relapsing–remitting bilateral encephalitis, Rasmussen encephalitis, encephalomalacia, brain biopsy, rituximab

## Abstract

A nine-year-old boy manifested with headache, progressive mild cognitive decline and hemiparesis, but without clinical epileptic seizures (with abnormal EEG waves). Brain magnetic resonance imaging (MRI) showed bilateral cortical lesions mainly on the right hemisphere, and new lesions developed in frontal, parietal, occipital and temporal lobes around the old lesions presenting as a lace-like or ring-like enhancement in T1 with contrast over a disease course of five years. A suspected diagnosis of primary angiitis of the central nervous system was initially considered. Treated with high-dose corticosteroids, intravenous immunoglobulins and monthly pulse cyclophosphamide, his symptoms worsened with the intracranial lesion progression. Brain biopsy of the right frontal lobe was performed nearly five years after onset; prominent neuronal loss, a microglial nodule, as well as parenchymal and perivascular lymphocytic infiltrate within the cortex were found, which coincided with RE pathology changes. Encouragingly, after a regimen of rituximab, lesions on the follow-up brain MRI tended to be stable. Apparently, it was immune-mediated, but did not strictly fit any known disease entity, although it was similar to RE. We summarize this unique case, including clinical characteristics, imaging and pathology findings. We also discuss the diagnosis and treatment, focusing on comparison to RE as well as other possible neurological diseases.

## 1. Introduction

Encephalitis means brain inflammation, which is usually caused by an infection or autoimmune inflammation; the latter can be roughly divided into antibody-mediated or cell-mediated. Immune-mediated encephalitis can be difficult to diagnosis due to its rarity and diversity. Rasmussen encephalitis (RE) is a rare immune-mediated (cell-mediated) neurological disease, and typically presents with progressive unihemispheric cortical atrophy [1,2]. It primarily occurs in adolescents or young adults with a prevalence of approximately 1.7 to 2.4 per 10 million people ≤ eighteen years of age [2,3]. It is characterized by intractable seizures, cognitive impairment, hemiparesis, hemianopia or aphasia. Epilepsia partialis continua was found in approximately 50% of patients with RE [4]. Robust imaging evidence reveals that unilateral perisylvian, temporal, frontal, parietal and occipital lobes are the most vulnerable regions of the central nervous system (CNS) [5]. In addition, unilateral movement disorders are less common presentations, including hemiathetosis and hemidystonia caused by the involvement of the lenticular nucleus and the caudate nucleus [6,7]. The course of the disease is classified using prodromal, acute and residual periods based on the severity and frequency of symptoms [2,8]. However, the presence of bilateral RE is rare and controversial. Rare cases of reported RE have been pathologically confirmed as definitive bilateral RE over the years [9,10]. In addition, there are cases with no seizures or delayed seizures during two years of follow-up, indicating that seizures are not obligatory for diagnosing this disease [11,12]. 

In this study, we present a special case of encephalitis with a relapsing–remitting disease course. The biopsy showed T-cell-mediated pathology changes as RE, although the clinical features did not include epilepsy and MRI showed bilateral involvement. We report and discuss this case, including the details of the diagnosis and treatment process, aiming to highlight the difficulties in differential diagnosis and the limitation of current RE diagnosis criteria.

## 2. Case Report

A nine-year-old boy initially presented with an episode of headache, nausea and vomiting six years ago when he was three years old. He is the only child in the family and his developmental milestones and mental status were normal. There is no family history of genetic disorder. His parents denied a history of fever or epilepsy during the course of his disease. Neurological deficits were not observed. The lumbar puncture pressure was 140 mm H_2_O. His initial brain MRI showed focal cortical encephalomalacia foci in bilateral frontal lobes, more severe on the right (Figure 1). Decreased values in apparent diffusion coefficient (ADC) and increased diffusion-weighted imaging (DWI) signals were observed over the affected hemispheres. In addition, the foci were restricted to focal lace-like enhancement in T1 with contrast (Figure 2) and showed decreased choline in magnetic resonance spectroscopy (MRS). Magnetic resonance angiography (MRA) and magnetic resonance venography (MRV) revealed no evidence of vascular abnormality or venous thrombosis. No vascular malformations or vasculitis were found on the digital subtraction angiography (DSA). No infectious, autoimmune or paraneoplastic markers were found. The presumptive diagnosis of bilateral nonspecific encephalitis was made based on the imaging. His parents refused consent for a stereotactic brain biopsy at first. The patient was treated with mannitol and glycerin fructose for decreasing intracranial pressure. His headache was fully relieved and he was discharged from hospital and returned to school.

The patient had undergone serial brain MRI scanning for regular evaluations of the intracranial lesions which showed progressive bilateral hemispheric atrophy and deterioration in various lobes. However, his condition did not appear to evolve and there was no clinical attack of headache, seizures or other neurological defects.

By August 2016, ten months after his first imaging, progressions were observed in a further brain MRI. The ring enhancing lesions around the encephalomalacia foci in the bilateral frontal lobes disappeared, which suggested gliosis.

Four years later, the patient was admitted in our hospital again because of a new episode of headache with nausea and vomiting. The neurologic examination showed intact cognitive function except a mild impairment of mathematical ability. The motor examination revealed a left hemiparesis, the left knee tendon reflex was active and the ankle clonus was positive. The right knee tendon reflex and Achilles tendon reflex were normal. The pathological reflexes were negative. The pressure of cerebrospinal fluid (CSF) was within normal range. CSF cell count and chemistry were normal. Metagenomic next-generation sequencing (mNGS) did not detect any pathogenic microorganisms in CSF. Antibodies against NMDAR, LGI1, AMPAR1, AMPAR2, GABAAR, GABABR, CASPR2, mGluR1, mGluR5, KLHL11, DRD2, D2R, Neurexin-3α, DPPX, IgLON5, GlyR-α1, AQP4, MOG, GFAP, Hu, Yo, Ri, CV2, Ma1, Ma2, SOX1, Zic4, GAD65, Tr/DNER, Titin, PKC-γ, recoverin, amphiphysin and IgG oligoclonal band were negative in both CSF and serum. Video electroencephalogram (VEEG) monitoring revealed entire slow waves with intermittent frontal, centrotemporal sharp and spike slow waves, which were dominant in the right hemisphere, and no periodic discharges over the bilateral hemispheres. The brain MRI revealed new lesions in the right frontal, parietal and temporal cortices and right cortex next to the cerebral falx (Figure 1). The range of encephalomalacia was enlarged on the primary basis and the corresponding blood perfusion in perfusion-weighted imaging (PWI) was decreased. The supratentorial ventricles and lateral fissure districts were obviously dilated. The involved areas showed hyperintensity in DWI and ADC hypointensity, but no defined hypointensity in susceptibility-weighted imaging (SWI). Diffusion tensor imaging (DTI) revealed asymmetric corticospinal tracts, and some tracts in partial scanning planes were compressed. The symptoms of the patient improved after treatment with mannitol for three days.

Five years later, new lesions in the right frontal and parietal cortices suggested the progression of the disease. MRS revealed decreased N-aspartic acid (NAA) in the involved lesions compared with the contralateral corresponding areas. No pathogenicity genes or copy number variants were found under the extensive evaluation of whole-genome sequencing, mitochondrial genes and copy number variants. Consent was obtained from the parents, and a brain biopsy in the right frontal lobe was eventually performed. The pathologic findings included neuronal loss, activated astrocytes, microglial nodules, parenchymal and perivascular T-lymphocytic infiltrate and neurotropic phenomenon within the cortex, which proved a cell-mediated etiology (Figure 3). The patient was therefore treated with mannitol and intravenous methylprednisolone pulse treatment, followed by oral prednisone at an initial dose of 30 mg daily with a slow tapering schedule of 5 mg every month. The latter immunotherapy strategies lasted for one-and-a-half years, and the detailed regimens were three pulses of intravenous methylprednisolone, three consecutive infusions of intravenous immunoglobulin (IVIg), and five pulses of 0.7 mg/m^2^ of cyclophosphamide monthly. The intracranial lesions were still progressing with relatively stable neurological deficits.

In March of 2021, the progressive lesions urged us to administer rituximab for maintenance treatment with an induction dosage of 375/m^2^ for three consecutive weeks. The maintenance rituximab treatments were infused based on the values of CD19, CD20 and CD19/CD27, and follow-up is still ongoing. Remarkably, the patient had a stabilized condition, and the brain lesions also tended to be stable nearly five months after infusions of rituximab (Figure 1). The patient presented no adverse or unanticipated events and his parents were satisfied with the treatments and the prognosis for their child. Further work on the treatment strategy is to continue to apply rituximab in the consolidation period. If relapse occurs, other immunosuppressants will be considered. The timeline of the patient with relevant data regarding episodes and interventions is presented in Figure 4.

## 3. Discussion

This case describes an actual example of bilateral relapsing–remitting encephalitis confirmed by serial brain MRI scanning, EEG topography and brain biopsy. During the course of the disease, the patient had several relapses of headache, nausea and vomiting, consequent hemiplegic paralysis and cognitive retardation rather than typical refractory epileptic seizures, continuous deteriorating neurological deficits and severe cognitive decline. The late confirmatory biopsy in the dominant affected hemisphere suggested microglial nodule, neuronal loss, perivascular and perivascular lymphocytic infiltrate within the cortex and deeper white matter, supporting a diagnosis of RE pathology. Clinical features without epilepsy and bilateral MRI changes makes a diagnosis of RE impossible, as RE criteria specify that seizures, focal neurological symptoms and MRI findings must be unilateral.

However, several bilateral RE cases have been reported in the past decades with evidence of EEG topography, brain MRI/CT or biopsy [9,10,13,14,15,16,17,18,19,20]. A review of the literature from 1993 to 2013 [9,10,13,14,15,16,17,18,19,20] identified that 16 patients have been reported with bilateral RE, all of whom had typical seizures, motor deficits or cognitive decline. Wallerian degeneration may play a role in the possible pathogenetic process of inflammation and encephalomalacia in the incidental hemicerebrum [1]. In a review of nine articles from 1991 to 2013, 87.5% (14/16) of patients were adolescents under the age of 18. It was found that disabilities in children were more severe and that the disease progressed more rapidly in children than in adults. In contrast, three adolescent patients had no seizures during the follow-up period of 1.3–1.9 years [12]. It was quite astonishing that our patient had no clinical epileptic seizures, but had abnormal EEG epileptic waves for six years. This finding indicates, as suggested by Olson et.al. in 2013 [21], that this RE diagnostic criterion should be updated; clinical epileptic seizures may not be an essential condition for RE, although its mechanism is not clear. Collectively, bilateral RE cases without clinical seizures would be another phenotype, which may be underestimated, easily leading to delayed or missed diagnoses.

Serial multi-model MRIs have been used to demonstrate bilateral relapsing–remitting encephalitis based on the appearance of new and old lesions, the features of the signal changes and atrophy degree in different stages [15,19,22]. Robust imaging evidence shows T1 hypointense and T2/FLAIR hyperintense signals in cortical and subcortical regions, which are the basic imaging characteristics. Although gadolinium enhancement is a diagnostic criterion for Rasmussen encephalitis according to the European consensus committee, very few reports mention details of the enhancement patterns [23]. As for our patient, the enhancement was restricted to focal lace-like and a ring in T1 with contrast once the new lesions occurred around the old lesions. The manifestation of brain DWI and ADC indicated relatively high molecular motion in the chronic inflammation period [24]. The imaging manifestations of new lesions in our patient also suggested vasogenic edema based on hyperintensities on DWI and the ADC map. MRS studies consistently showed decreased NAA levels and choline levels resulting in a decreased NAA/choline ratio, which suggested neuronal loss or dysfunction [5]. In addition, diffuse hypometabolism was found using fluorodeoxyglucose positron emission tomography (FDG-PET), indicating the involvement of the unilateral hemisphere when MRI atrophy was not evident [25]. DTI also revealed asymmetric corticospinal tracts and PWI demonstrated decreased brain blood perfusion. In the literature review, more than 56% (9/16) of patients were diagnosed with bilateral RE based on a brain MRI or CT scanning. However, such information is apparently vague and general, without additional details regarding the progression of brain lesions. We analyzed and evaluated brain lesions using serial multi-model MRIs for more than six years, which expanded the clinical profile of the disease and helped to understand and monitor the disease progression.

In the early stage of the disease, because a brain biopsy had not yet been obtained, a definite diagnosis of the patient could not be established. Particular attention was paid to the differential diagnoses of bilateral cortical/subcortical encephalopathy. The acute onset in childhood of a relapsing–remitting disease course led us to consider several possible diseases [26]. First, CNS infectious diseases were considered; however, the patient did not have fever, mental or behavior disorders, disturbance of consciousness, or epileptic seizure. Additionally, tests for serum syphilis and treponemal antibodies, HIV, CSF viral PCR including HSV, VZV, HHV6, JCV, and CSF VZV IgG/IgM had negative results. Second, CNS demyelinating diseases, such as acute disseminated encephalomyelitis, acute hemorrhagic leukoencephalitis, tumefactive disseminated sclerosis, Marburg, MOG antibody-associated disease, and neuromyelitis optica spectrum disorders might share many characteristics and were difficult to identify in brain imaging or even in serial brain MRIs during long-term follow-up. However, antibodies for MOG and AQP4 were negative, diverse immunomodulatory therapies did not stop the spread of the brain lesions, especially in the unilateral cerebral hemisphere, and the late brain biopsy excluded CNS neoplasms. Furthermore, the later brain biopsy showed neuronal loss, parenchymal and perivascular lymphocytic infiltrate and microglial nodules, and thus excluded the suspected intracranial tumor (such as lymphoma or glioma) and primary angiitis of the central nervous system (PACNS, because granulomatous angiitis, lymphocytic vasculitis and even necrotizing angiitis were all absent) [27]. Other systemic autoimmune diseases, such as lupus cerebritis, Behcet disease and neurosarcoidosis, were also excluded, as these diseases might manifest as bilateral cortical/subcortical encephalitis. Hereditary diseases were the inevitable challenge to accept, and serum whole genome sequencing did not suggest hereditary leukoencephalopathy. In a previous study, a variety of single nucleotide variants associated with antigen presentation, epilepsy, schizophrenia and nerve cell regeneration was found in RE brain biopsy tissue using a whole genome sequencing method, which suggested polygenic harmful factors’ involvement [28]. Although not performed in the current study, genetic or transcriptomic studies with brain biopsy tissues should be taken into consideration in this type of patient to deepen our understanding of the pathological mechanism of RE. Other toxic, traumatic, endocrine and metabolic disorders and cerebrovascular diseases were eliminated due to the absence of exposure of the corresponding risks. Collectively, the combination of relapsing–remitting symptoms, serial brain MRI scanning, EEG topography and brain biopsy implied two of three items in Part B of Bein’s criteria for the diagnosis of RE, after the exclusion of other etiologies [21].

The pathogenesis of RE remains largely unclear and treatments for RE are still rather controversial. To date, there are no standard recommendations or consensus-based practice guidelines for the treatment of RE because the mechanisms of this disease are still unclear. The admitted aim of RE treatment is to reduce episodes of seizures and improve the long-term outcomes of neurological function and cognitive performance [2]. Anticonvulsive pharmacotherapy, immunosuppressants and hemispherectomy are alternatives to reduce functional and structural damage. Hemispherectomy was not applicable in this case, as the patient had no refractory epilepsy, although the appearance of contralateral EEG abnormalities may highlight the risk of disease deterioration. In view of the immunopathological basis of RE, antibody-mediated, T-cell cytotoxicity- and microglia-induced degeneration may lead to progressive immune-mediated neuronal destruction [1,2]. Further clues to an immune-mediated mechanism are the discovery of the potentially pathogenic antibody against GluR3, alpha-7 nicotinic acetylcholine receptor or Munc-18-1, although these antibodies were found in few patients with RE [2]. Therefore, various approaches, including immunosuppressive or immunomodulatory treatments, targeted to different steps of the pathogenesis, are applied empirically. Steroid pulse therapy, intravenous immunoglobulins, plasmapheresis or immunoabsorption, T-cell-inactivating drugs such as tacrolimus and azathioprine, and indirect B-cell-reducing drugs such as rituximab have been administered in a few cases [29]. However, large numbers of patients with RE need to be identified to prove the effectiveness of each drug. As for our patient, diverse immunomodulatory therapies were administered from the beginning of treatment; however, intracranial lesions became inexorable within both the cortex and subcortical white matter. After full communication with the parents, the patient accepted infusions of rituximab and achieved a relatively stable condition. In fact, in 36 cases, rituximab was well tolerated and patients benefited from improvement in seizures, and motor and cognitive function [30]. Therefore, rituximab could be a candidate for the treatment of RE before undergoing a hemispherectomy [29].

## 4. Conclusions

Bilateral involvements with progressive lesions may serve as the indispensable pattern of RE in the relapsing–remitting disease course, if under the condition with a broadened diagnostic criteria. Serial multi-mode brain MRI scanning and brain biopsy are crucial for a definite diagnosis as well as monitoring disease progression in the longitudinal follow-up period. Development and application of a new type of immunosuppressant, rituximab, may be an alternative therapy to help prevent RE progression and decrease associated disability progression.

## Figures and Tables

**Figure 1 brainsci-13-00017-f001:**
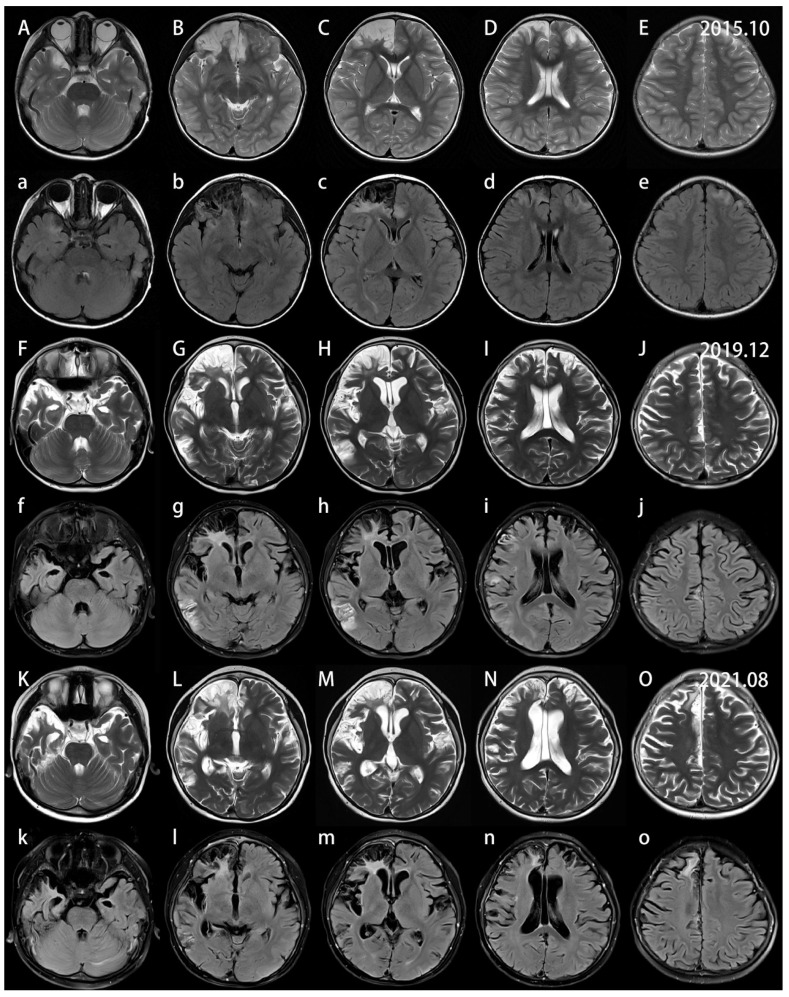
Control brain magnetic resonance imaging (MRI) of the patient at the initial onset of the disease in October 2015, palindromia in December 2019 and stable period after rituximab in August 2021. Axial T2-weighted (**A**–**O**) images and axial FLAIR-weighted (**a**–**o**) images show prominently progressive cortical and subcortical atrophy and encephalomalacia in the right frontal, temporal and insula lobes, and slowly progressive focal atrophy in the contralateral frontal and insular lobes. The new patchy lesions around the old lesions are revealed clearly on axial FLAIR-weighted (**a**–**o**) images. The bilateral lateral ventricles and lateral fissure districts dilated, particularly as the disease progressed in December 2019 (**F**–**J**,**f**–**j**), but the lesions seemed to be under control to some extent with rituximab treatment (**K**–**O**,**k**–**o**).

**Figure 2 brainsci-13-00017-f002:**
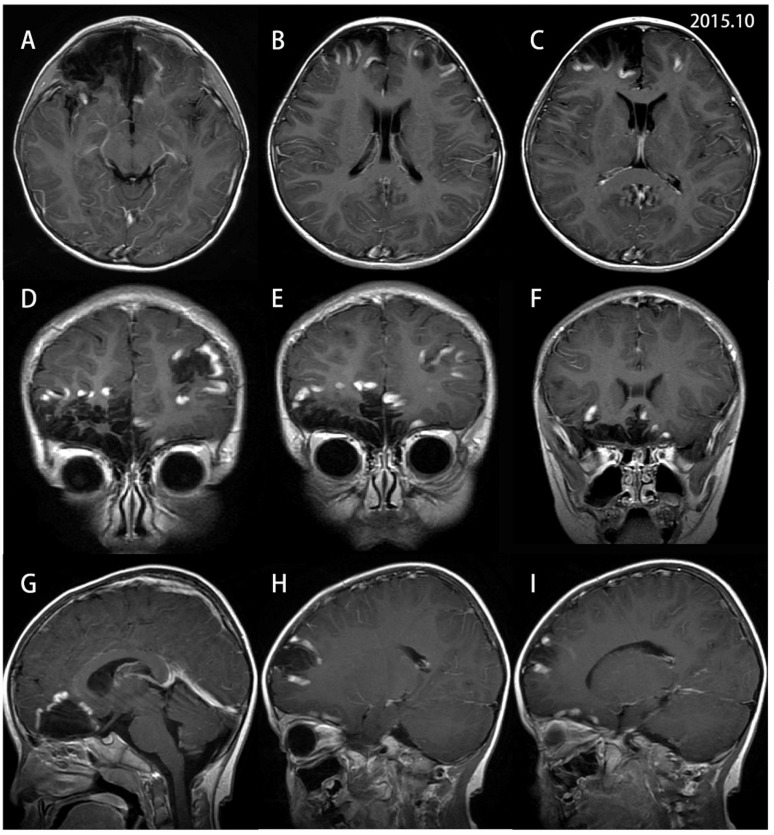
Gd-enhanced T1-weighted MRIs of the patient were performed during the initial acute attack in October 2015. Axial (**A**–**C**), coronal (**D**–**F**) and sagittal (**G**–**I**) images show multiple enhanced lesions restricted around the previous lesions or districts of encephalomalacia, which were revealed with lace-like enhancement.

**Figure 3 brainsci-13-00017-f003:**
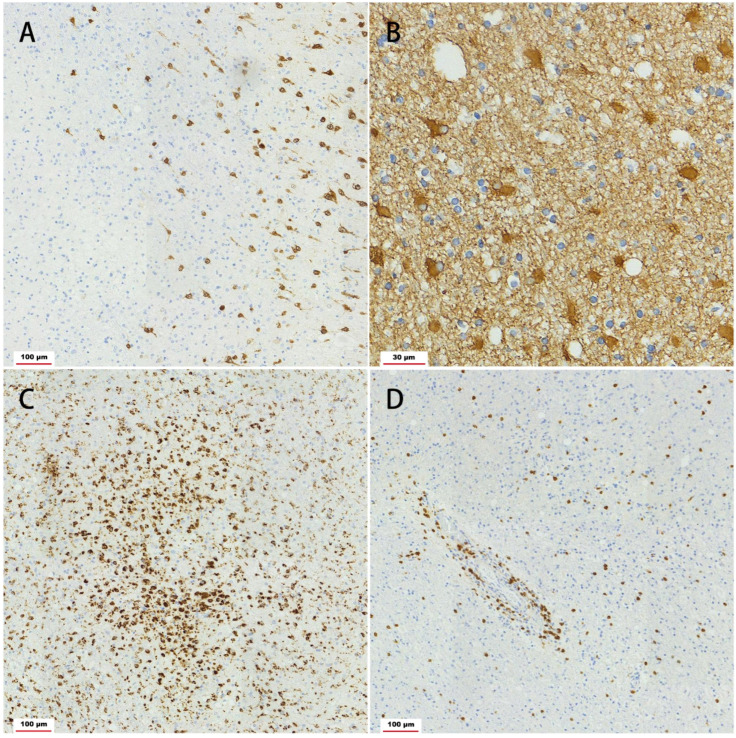
Light microscopy of the brain biopsy. Staining for neuronal nuclear antigen showed the loss of cortical neurons in the left side, whereas intact neurons were found in the right side ((**A**), Scale Bar = 100 μm). The same areas were stained for glial fibrillary acidic protein, showing strongly activated astrocytes ((**B**), Scale Bar = 30 μm). CD68 staining revealed the presence of microglial nodules in the cerebral cortex ((**C**), Scale Bar = 100 μm). CD3 staining revealed the parenchymal and perivascular cuffing with T lymphocyte ((**D**), Scale Bar = 100 μm).

**Figure 4 brainsci-13-00017-f004:**
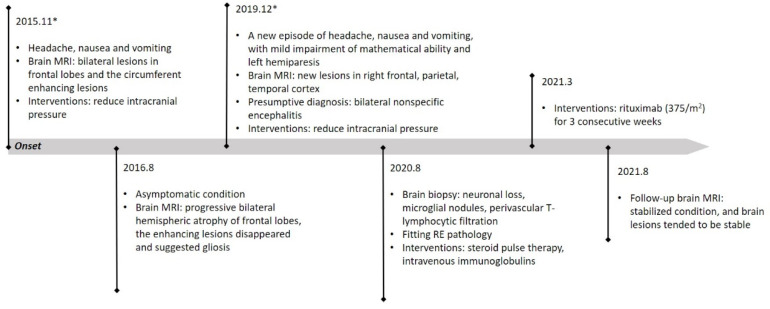
The timeline of the patient with relevant data regarding past episodes and interventions. *: two admissions.

## Data Availability

Raw data supporting the conclusions of this article will be made available by the authors, without undue reservation.

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
