# Peer review of "A Special Case of Relapsing–Remitting Bilateral Encephalitis: Without Epilepsy, but Responding to Rituximab and with a Brain Biopsy Coinciding with Rasmussen Encephalitis"

_brainsci, 2022, doi:10.3390/brainsci13010017_

Round 1

Reviewer 1 Report

Dear authors

The authors kindly submitted a case report, entitled "Case Report: Radiographic Progression in A Bilateral Rasmussen Encephalitis." The authors demonstrated one 9-year-old boy of bilateral Rasmussen Encephalitis (RE) with unique clinical features, the diagnostic process, treatments and outcome, with the focus on the serials of brain MRI scanning performed to evaluate the lesion progression during the follow-up period.

The writing is good and logical. However, due to the rarity, the authors should present the case report with more scientific values, showing new insights, new treatment experience, new diagnostic methods, or genetic findings.  In addition, there are also minor parts essential to be revised. The followings are the comments:

1) The image data in Figure 2 were not well labelled/introduced and explained both in the main text and in the figure legend. The authors didn't explain the radiological modalities (e.g. T1FLAIR with contrast, T2FSE, DWI....). Moreover, the authors didn't explain the findings in detail.

2) The supplement file is the CVs of all authors. This is not what supplementary data stand for. The supplement files should be the additional materials and results which have no room put in the draft. 

3) Although the patient consensus was said to be received, the authors didn't complete the Institutional Review Board Statement. Please complete it, especially the patient is only 9 years of age. 

4) Since the brain biopsy was done, the genetic or transcriptomic studies (e.g. WES, RNAseq) should be done with brain tissues, not with blood, and further illustrate the possible pathological mechanism.  In particular, other articles have reported some genetic alteration (https://www.frontiersin.org/articles/10.3389/fnins.2021.744429/full). 

5) In Figure 3, the bar in the B panel showed 20μm, but the figure legend said 30μm. Please correct the inconsistency. 

6) There is no conclusion part.

Reviewer 2 Report

I agree with the authors that  the case report is extremelly rare and at the same time controversial. It is such an unsual presentation that the diagnosis of RE(supported mainly by the brain biopsy) is uncertain. The clinical presentation (without seizures or even progressive neurological deficits) is atypical, the clinical course (relapsing-remiting) and  the MRI (bilateral but also with encephalomalacia areas and without caudate and putamen atrophy and witthout affecting perisylvian region predominantly).  For all this atypical characteristics the case does not fullfil the Bien diagnostic criteria , part A or B .The Criteria speciffy that seizures, focal neurological symptoms and MRI findings must be unilateral. In this situation it is extremely important to discard all other possible diagnosis (specially viral /fungal infeccions). I suggest to add a table including all the diagnostic workup done and specifying the possible ruled out aetiologies

Round 2

Reviewer 1 Report

Dear authors

The authors kindly submitted a revised version of case report, entitled "Case Report: Radiographic Progression in A Bilateral Rasmussen Encephalitis." They demonstrated one 9-year-old boy of bilateral Rasmussen Encephalitis (RE) with unique clinical features, the diagnostic process, treatments and outcome, with the focus on the serials of brain MRI scanning.

It's a pity that the case report doesn't provide any new insights, new experience of clinical trial, or the patient's genetic findings, although a rare case with educational values may be acceptable. However, the authors have revised my points well. I have no further comments on the current version. 

Author Response

Thanks for your kind consideration.

Reviewer 2 Report

Dear authors 

I repeat  that the case does not fullfil the diagnostic criteria for Rassmussen encephalitis (part A or part B). The clinical presentation (without seizures or even progressive neurological deficits) is atypical, the clinical course (relapsing-remitting) and the MRI (bilateral but also with encephalomalacia areas and without caudate and putamen atrophy and without affecting perisylvian region predominantly). The criteria specify that seizures, focal neurological symptoms and MRI findings must be unilateral. In this situation you cannot diagnose the patients as a atypical RE ,I suggest to define it as a relapsing- remiting  bilateral encephalitis.  

  On the other hand  I am not satisfied woth your differential diagnosis work up. Metagenomic next-generations sequencing does not discard all CNS infectious diseases and  negative whole-genome sequencing does not discard all genetic /metabolic diseases.

Please specify which diseases have been excluded ( virus herpes simplex  encephalitis by metagenomic next-generation sequencing (for example).

I include a table of  possible diferential diagnosis if this can help   

Author Response

Thanks for your kind comments.
